# Determination of the Viability of Lactic Acid Bacteria by Dynamic In Vitro Gastrointestinal Model in Household and Industrial-Type Kefir Samples

**DOI:** 10.3390/nu15224808

**Published:** 2023-11-17

**Authors:** Merve İnce Palamutoğlu, Gizem Köse, Murat Baş

**Affiliations:** 1Department of Nutrition and Dietetics, Institute of Health Sciences, Acibadem Mehmet Ali Aydinlar University, Istanbul 34752, Turkey; 2Department of Nutrition and Dietetics, Faculty of Health Sciences, Afyonkarahisar Health Sciences University, Afyonkarahisar 03030, Turkey; 3Department of Nutrition and Dietetics, Faculty of Health Sciences, Acibadem Mehmet Ali Aydinlar University, Istanbul 34752, Turkey; gizem.kose@acibadem.edu.tr (G.K.); murat.bas@acibadem.edu.tr (M.B.)

**Keywords:** dynamic in vitro model, fermentation, in vitro gastrointestinal model, probiotic, kefir

## Abstract

This study presents results based on differences in the antioxidant activity and lactic acid bacteria counts in different parts of the digestive tract following simulated gastrointestinal digestion of kefir samples. Statistically significant differences were observed in Lactobacillus counts in different kefir types including industrial (IK), starter culture (SCK), and kefir grains (KG). These differences were observed between the initial and second min in the mouth region (T = 3.968; *p* < 0.05); and between the initial, 60th, and 120th min in the stomach region (R = 11.146; *p* < 0.05). Additionally, a statistically significant difference was noted in the initial Lactobacillus levels among the IK, SCK, and KG in the stomach region (H = 7.205; *p* < 0.05). Also, significant differences were identified between the Lactococcus counts of IK across 0, 60, and 120 min in the stomach region (R = 10.236; *p* < 0.05). Notably, a statistically significant difference was noted in the Lactococcus levels in the KG between the initial and second min in the mouth region (T = 3.101; *p* < 0.05) and between 0, 60, and 120 min in the stomach region (R = 25.771; *p* < 0.001). These findings highlight the differences between the physicochemical characteristics of different kefir types. A decrease in lactic acid bacteria counts in kefir samples was observed throughout the dynamic in vitro gastrointestinal tract to reveal the significance of the digestive process when determining probiotic product capacity.

## 1. Introduction

The Food and Agriculture Organization (FAO) and the World Health Organization (WHO) recently defined probiotic bacteria as “live microorganisms that, when consumed in adequate numbers, confer a beneficial health effect on the gastrointestinal tract of the host” [1,2]. To provide these benefits, probiotic bacteria should be at a minimum level of 10^6^ colony-forming units per g or mL (cfu/g or mL) upon reaching the intestines. Risk factors for the viability of probiotic bacteria include various stress factors during production, storage, and passage through the gastrointestinal tract [2]. The gut microbiota has been widely implicated in several host diseases. One of the most significant strategies for modulating the gut microbiota for human health is sufficient dietary intake of probiotics [1]. *Lactobacillus* and *Bifidobacterium* species are among the most widely used probiotic bacteria owing to their beneficial effects on human health. However, yeasts such as *Saccharomyces boulardii* and *Saccharomyces cerevisiae* are also used as probiotics for their beneficial effects. To date, the ability of probiotics to modulate certain functions in the body has led to their widespread use in the manufacture of food, beverages, and supplements. However, frequent reports of probiotic formulations with inferior microbiological quality in the market are noted, as determined by the identification and viability of the microorganisms they include. These qualitative defects may stem from the implementation of subpar or insufficiently rigorous procedures by researchers, as well as insufficient quality controls by manufacturers. The European Society for Pediatric Gastroenterology, Hepatology, and Nutrition has underlined the significance of stricter control of commercialized probiotic products. Several organizations worldwide are focused on the quality control of probiotic products. Therefore, existing and future probiotic formulations must satisfy the essential criteria of qualitative and quantitative compatibilities between what the label claims and what the formulation contains. In fact, probiotic product labels must specify the minimum number of viable cells to remain in the product until the end of its shelf life as well as the names of all the microorganisms it contains following the guidelines for probiotics issued by the FAO of the United Nations, the WHO, the International Society for Probiotics, and the Council for Responsible Nutrition. Probiotic products containing enough living organisms must be used to obtain the desired beneficial health effect. This quantity is generated from in vitro and in vivo testing and depends on the stability of the species and strains contained; thus, it is not consistently calculated for all goods [3]. Probiotics must be resistant to gastric juice and bile salts to reach the intestinal environment intact and exert their beneficial activities on the host organism. The large intestine is the site of action for most probiotic bacteria. Losses may occur throughout the gastrointestinal tract; however, the acidic environment of the stomach and the presence of bile in the duodenum are the main factors affecting the viability of probiotic bacteria. Viability during production and shelf life, as well as the type and oxygen permeability of the food packaging, are other factors to consider. In vivo studies are highly complex for use in the initial screening experiments necessary for selecting potential probiotic bacteria and examining their behavior in food matrices [4].

Kefir is the most interesting beverage, which has a complex structure and probiotic microbes. Traditional or commercial semi-skimmed or skim milk (from goats, cows, sheep, and camels) is fermented using kefir grains to create kefir [5]. Kefir has a slightly acidic refreshing taste owing to its lactic acid content [6]. It is known for its easy digestibility and higher nutritional value. Different kefir microflora exist depending on the kind of kefir grain, culture media, and production technique. Intricate interactions between the kind and quantity of milk, yeast, and lactic acid bacteria can affect the sensory and textural characteristics of kefir [7]. The lactic acid bacteria in kefir are added to milk to help start fermentation. These microorganisms convert lactose into lactic acid, which reduces the pH level. Some of the fundamental characteristics of fermented milk including taste and long shelf life are brought about by lactic acid bacteria [5]. The color, taste, odor, chemical composition, and microbial content of kefir may vary depending on the diversity in the microflora of kefir grains, the number of microorganism species and populations, the biochemical properties and microbiological profile of the milk used, or the production method [8].

In vivo studies on humans are the most ideal for obtaining reliable and accurate data on the determination of microorganism viability in gastrointestinal conditions. However, human clinical studies are challenging as they are not frequently technically, financially, and ethically viable, and they are associated with low reproducibility due to individual differences [9]. Owing to the complicated processes that take place during human or animal digestion, and the technically difficult, costly, and, in particular, ethical constraints, in vivo research is exceedingly challenging [10]. Therefore, considering various factors including the formation and concentration of digestive enzymes, gastric and intestinal stages, digestion time, and pH values has led to the design and use of reliable in vitro models that can closely mimic the human gastrointestinal tract. In vitro system models offer a useful alternative to human and animal models because they are flexible, accurate, and reproducible [11]. In vitro gastrointestinal models are divided into the following two types: static and dynamic. In static models, the foodstuff is exposed to the processes in the digestive system, and time-dependent parameters (pH, enzyme level, and sample amount) are not considered during digestion, whereas in dynamic models, time-dependent parameters are simulated during digestion by physical and mechanical processes [10]. Depending on the purpose of the study, the simulated digestion model may include the mouth, stomach, and small intestine stages and, in some cases, colonic fermentation. The simulated mouth, stomach, and small intestine are incubated with gastric and digestive juices for a set amount of time and temperature, while the pH is kept constant at each stage using a suitable buffer solution [12].

This study aimed to compare the proximate composition of the kefir samples, antioxidant activity, and number of lactic acid bacteria with the initial probiotic character and examine the changes throughout the simulated gastrointestinal tract.

## 2. Materials and Methods

### 2.1. Materials

The kefir used in this study consisted of 12 different samples, including 5 home-type kefir samples produced using kefir grains in the laboratory, 2 kefir samples produced in the laboratory using starter culture powder, and 5 industrially produced plain kefir samples purchased from the market. The same brand and batch number of pasteurized cow milk were used to produce the kefir samples.

Kefir production using kefir grains in a laboratory environment was performed by directly adding kefir grains to pasteurized milk and leaving it to ferment for 24 h in a dark environment at room temperature. In the production using starter culture powder in a laboratory environment, kefir was added to pasteurized milk following the production instructions on the product label. Plain kefir samples purchased from the market were obtained from manufacturers with different brands without specifying the brand and trade names.

In this study, alpha-amylase, mucin, bile salt, pancreatin, and pepsin were obtained from Sigma-Aldrich (St. Louis, MO, USA); CaCl_2_, NaCl, NaHCO_3_, and NaOH were obtained from AFG Bioscience (Northbrook, IL, USA); KCl was obtained from TEKKİM (Bursa, Turkey), and HCl was obtained from Honeywell (Seelze, Germany). Ringer tablet and MRS agar medium were obtained from Neogen Culture Media (Lansing, MI, USA), and M17 agar was obtained from Condalab Laboratorios (Madrid, Spain). Disposable sterile petri dishes were obtained from Fıratmed (Ankara, Turkey). The anaerobic conditioning reagents Anaerobentopf 2.5 l-Volumen, Microbiology Anaerocult A, and Anaerotest strips (Merck KGaA, Darmstadt, Germany) were used.

### 2.2. Development of the Dynamic In Vitro Gastrointestinal Model

The model consists of three consecutive sections that represent the mouth, stomach, and small intestine regions. A temperature-controlled water bath was used to represent the mouth region, and two double-jacketed reaction vessels kept at 37 °C ± 0.1 °C were used to represent the stomach and small intestine regions. Instant temperature and pH monitoring were performed. Dynamic in vitro gastrointestinal transit time was kept under control for 2 min, 2 h, and 2 h in the mouth, stomach, and small intestine regions, respectively. Peristaltic pumps with adjustable speeds were used to control the flow of the digestive system secretions to be simulated and ensure the passage of the sample, which should be exposed to the processes in the digestive system, from the mouth to the stomach and from the stomach to the small intestine. Additionally, to keep the pH level of the stomach and small intestine regions constant, pH balance was achieved using 1 M NaOH and 0.2 M HCl when necessary, with instant pH monitoring.

Following the entrance of the samples into the mouth of the developed dynamic in vitro gastrointestinal model, salivary secretion was added. The salivary secretion was mixed at 20 °C by adding 2 g/L α-amylase enzyme with a pH level of 6.9, 1 g/L mucin, 25 mL 0.3 M CaCl_2_, and 975 mL water; the pH level of the medium was adjusted for 2 min and kept constant throughout. A 0.05 mL/g simulated saliva sample was added to the mouth environment at a 5 mL/min rate. The contact time with the enzyme was adjusted to be 2 min at 37 °C. Regarding the gastric buffer solution, it was prepared using 2.2 g/L KCl, 6.2 g/L NaCl, 1.2 g/L NaHCO_3_, and 0.22 g/L CaCl_2_. In the stomach region, to simulate gastric secretion, 3700 ppm/L of pepsin enzyme and 23 g/L of mucin were dissolved in a sterile gastric buffer solution. The simulated gastric secretion was added to the reactor representing the stomach at a 0.25 mL/min rate, as a 0.05 mL gastric secretion/g sample. After the sample left the mouth environment, underwent digestion, and passed into the reactor representing the stomach environment with a 100 mL/min flow rate, 0.2 M HCl acid was added at a 3.5 mL/min rate until the pH level of the gastric reactor reached 2.5. After a gradual decrease to a pH level of 2.5 within 1 h, the transit rate of HCl acid was provided at 0.9 mL/min to simulate gastrin inhibition to keep the pH constant at 2.5 for 1 h. The double-jacketed reaction vessel was integrated into the circulating water bath, and the temperature remained constant at 37 °C for 2 h. Regarding the small intestine buffer solution, 0.6 g/L KCl, 5.0 g/L NaCl, 0.25 g/L CaCl_2_, 1 g/L pancreatin, 12 g/L bile salt, and 1 M NaHCO_3_ solution were dissolved in the small intestine buffer solution to simulate the secretion of the small intestine. The simulated small intestine secretion was added to the reactor representing the small intestine at a 0.25 mL/min rate, as a 0.05 mL small intestine secretion/g sample. After the sample left the stomach environment and underwent digestion, it passed into the reactor representing the small intestine environment with a 100 mL/min flow rate for 15–20 min. Subsequently, to gradually increase the pH level of the environment to 6.5, 1 M NaOH was added at a 0.65 mL/min transition rate. It was ensured that the pH level remained constant at 6.5 until the end of the digestion process. The double-jacketed reaction vessel was integrated into the circulating water bath, and the temperature remained constant at 37 °C for 2 h [10].

The temperature values and times applied during the digestion process and the composition, concentrations, and flow rates of saliva, stomach, and small intestine secretions and buffer solutions used were from gastrointestinal model studies obtained from the literature [4,10,12,13].

### 2.3. Proximate Analysis

The dry matter content was predried in an oven and cooled in a desiccator, and the amount of dry matter was calculated by weighing sensitively [14]. The fat content was determined using the Gerber method [15]. To determine the protein content of the samples, the Kjeldahl method was used [16]. Soluble Solids Content (°Brix) was measured using a hand refractometer (Fukui, Japan) at 20 °C. The pH levels of the samples were measured using an Inolab pH meter (Mettler Toledo, Columbus, OH, USA). Regarding titration acidity (% lactic acid), a certain amount of sample was titrated with 0.25 N NaOH solution in the presence of phenolphthalein indicator, and the result was calculated in terms of lactic acid [17].

### 2.4. 2,2-Difenil-1-Picrylhydrazyl (DPPH) Activity Determination

Before the analysis began, a 0.02 mM DPPH solution was prepared using methanol. Two hundred microliters of the samples were taken, whereas 600 μL of DPPH was added and incubated in Eppendorf tubes for 30 min in a dark environment at room temperature. The same procedure was followed using distilled water instead of the sample, and the blank was measured at 571 nm [18].

### 2.5. Microbiological Analysis

The lactic acid bacteria count in kefir samples was tested in the mouth for 0 min just before being introduced into the dynamic in vitro gastrointestinal model. While the kefir samples passed through the mouth, stomach, and small intestine of the dynamic in vitro gastrointestinal model, microbial samples were taken at the given times. Subsequently, 1 mL samples were taken from the mouth at 0 and 2 min, and from the stomach and small intestine at 0, 60, and 120 min. The number of live microorganisms in the samples was determined as cfu/mL. Before microbiological cultivations, suitable decimal serial dilutions were prepared under aseptic conditions using quarter-strength Ringer’s solution. For dilution, a 10^−1^ (1/10) dilution was prepared using 90 mL of sterile dilution solution in 10 mL of kefir from each sample. This process was repeated until the values of 10^−6^ and 10^−7^ were reached [10].

### 2.6. Lactobacillus and Lactococcus Count

Following dilution, MRS agar medium with a pH level adjusted to 5.2 was used for growth, and after at least 48 h incubation, viable cell counting of *Lactobacillus* cultures in kefir samples was performed [10,19]. M17 agar medium with 1% lactose added was used for the growth and enumeration of *Lactococcus* cultures [20]. One microliter was taken from the dilutions and poured onto MRS agar and M17 agar plates. The pour plate method was used for the analysis. After incubation was performed in anaerobic conditions at 37 °C for 72 h, lactic acid bacteria were counted using a colony-counting device [21]. Microbiological analyses of the kefir samples were performed using a colony-counting device and culture-counting methods. The determined bacterial numbers were calculated as cfu using the cfu/mL formula [22]. Colonies larger than 0.5 mm that developed on the medium at the end of the incubation were counted [23].

### 2.7. Statistical Analysis

The conformity of numerical variables to normal distribution was evaluated using the Shapiro–Wilk test. Descriptive statistics of numerical variables were expressed as means ± standard deviations ($\overset{-}{X}$ ± SDs) for normally distributed data and median (minimum–maximum) values for non-normally distributed data. The one-way analysis of variance (ANOVA) test was used to compare more than two independent groups with normal distribution, whereas the Kruskal–Wallis H test was used to compare more than two independent groups without normal distribution. The dependent samples *t*-test was used to compare two dependent times with normal distribution, whereas the repeated measures ANOVA test was used to compare two times with normal distribution. Moreover, to compare two dependent times with non-normal distribution, the Wilcoxon signed-rank test was used, and the Friedman test was used to compare more than two times. In this study, the statistical significance level was considered “*p* < 0.05, *p* < 0.01, and *p* < 0.001” in all calculations and interpretations, and the hypotheses were established bidirectionally.

All statistical analyses were performed using Statistical Package for the Social Sciences (version 26, IBM Inc., Chicago, IL, USA). The number of viable bacteria in one of the five different commercial products picked dramatically decreased at 120 min in the stomach, and the same level was maintained until the end of 120 min in the small intestine region. It was eliminated from the statistical analysis because it created conflicts, unlike other industrial-type production brands, and frequently generated a difference in comparison.

## 3. Results

The kefir analysis values of industrial kefir (IK), kefir produced using a starter culture powder (SCK), and kefir produced using kefir grains (KG) were compared. It was determined that for “dry matter” (H = 29.988; *p* < 0.001), “protein” (H = 26.840; *p* < 0.001), “titration acidity” (H = 18.058; *p* < 0.001), and “°Brix” (H = 32.583; *p* < 0.001), a statistically significant difference (*p* < 0.001) was noted between the values. No significant difference was noted between the ash and lipid contents of the kefir samples (Table 1).

No statistically significant difference (*p* > 0.05) was observed between the regional DPPH difference values according to the IK, SCK, and KG groups (Figure 1). In the KG stomach region, a statistically significant difference (R = 7.765; *p* < 0.05) was noted between the DPPH values at 0, 60, and 120 min; however, no significant difference was observed between the DPPH values in the mouth and small intestine regions (*p* > 0.05). The mean DPPH value at 0 min (54.48 ± 15.18) and the mean of the 60-min DPPH value (36.00 ± 6.11) in the KG stomach region were statistically higher than the mean of the 120-min DPPH value (25.41 ± 10.07). Compared with the kefir groups produced using IK, SCK, and KG, a statistically significant difference was noted between the DPPH values at 60 (H = 7.064; *p* < 0.05) and 120 min (H = 7.064; *p* < 0.05) in the stomach. Simultaneously, a statistically significant difference was observed between the small intestinal DPPH values at 0 (H = 7000; *p* < 0.05), 60 (H = 8.591; *p* < 0.05), and 120 min (H = 7.255; *p* < 0.05) (Table 2).

The comparison between the DPPH, *Lactobacillus*, and *Lactococcus* difference values in the mouth, stomach, and small intestine regions of the kefir groups produced using IK, SCK, and KG is presented in Table 3. A statistically significant difference was noted between the DPPH difference values in the stomach region (H = 7.573; *p* < 0.05) according to the kefir groups produced using IK, SCK, and KG, whereas no significant difference was observed between the mouth and small intestine region difference values (*p* > 0.05). Furthermore, no statistically significant difference (*p* > 0.05) was noted between the *Lactobacillus* difference values in the mouth, stomach, and small intestine regions compared with the kefir groups produced using IK, SCK, and KG. A statistically significant difference was observed between the regional *Lactobacillus* difference values (F = 4.840; *p* < 0.05) according to the groups produced using KG. In other words, the mean *Lactobacillus* difference value in the mouth region (−0.06 ± 0.03) in the KG group was statistically higher than that in the stomach region (−1.44 ± 0.59). No statistically significant difference (*p* > 0.05) was noted between the *Lactococcus* difference values in the mouth, stomach, and small intestine regions compared with the kefir groups produced using IK, SCK, and KG. A statistically significant difference was noted between the regional *Lactococcus* difference values in the KG groups (F = 4.151; *p* < 0.05).

The distribution of IK, SCK, and KG values in the kefir grain group included in this study, as well as the *Lactobacillus* values in the mouth region at 0 and 2 min, stomach and small intestine regions at 0, 60, and 120 min are shown in Figure 2. No statistically significant difference (*p* > 0.05) was observed between the *Lactobacillus* values at 0 and 2 min in the mouth region of IK and SCK kefirs and at 0, 60, and 120 min in the stomach and small intestine regions. A statistically significant difference was observed between the mouth (T = 3.968; *p* < 0.05) and stomach region *Lactobacillus* values in the KG groups (R = 11.146; *p* < 0.05); however, no significant difference was noted between the small intestine region *Lactobacillus* values (*p* > 0.05). A statistically significant difference was noted between the 0-min *Lactobacillus* values in the stomach region (H = 7.205; *p* < 0.05) according to the SCK and KG groups of IK. The stomach region *Lactobacillus* values at 0 min were statistically higher than the median of the IK (9.41 [8.90–9.69]), KG (8.62 [7.74–8.95]), and SCK (7.01 [6.11–7.90]) (Table 4).

The distribution of IK, SCK, and KG values in the kefir grain group included in this study, as well as the *Lactococcus* values in the mouth region at 0 and 2 min and the stomach and small intestine regions at 0, 60, and 120 min are depicted in Figure 3. A statistically significant difference (R = 10.236; *p* < 0.05) was noted between the *Lactococcus* values in the stomach region of IK; however, no significant difference was observed between the mouth and small intestine region *Lactococcus* values (*p* > 0.05). The mean *Lactococcus* value (10.01 ± 0.27) at 0 min in the stomach region of IK was statistically higher than that of the 120-min *Lactococcus* value (9.21 ± 0.24). No statistically significant difference (*p* > 0.05) was observed between the *Lactococcus* values in the mouth, stomach, and small intestine regions of the SCK groups. A statistically significant difference was noted between the mouth (T = 3.101; *p* < 0.05) and stomach region *Lactococcus* values (R = 25,771; *p* < 0.001) of the KG groups; however, no significant difference was noted between the intestinal region *Lactococcus* values (*p* > 0.05). In the IK, SCK, and KG groups, compared with the mouth at 0 (H = 6.505; *p* < 0.05) and 2 min (H = 8.591; *p* < 0.05), stomach at 0 min (H = 7.814; *p* < 0.05) and small intestine, a statistically significant difference was observed between *Lactococcus* values at 60 min (H = 6.505; *p* < 0.05) (Table 5).

## 4. Discussion

Preserving the viability of probiotic bacteria is essential for the effectiveness of probiotic products. Temperature, fermentation, and time of storage are factors to consider. Bacterial cell viability during production and shelf life, as well as the kind and oxygen permeability of the food packaging, are additional factors to consider. Probiotic bacteria must reach the colon in sufficient numbers and there must be at least 10^6^ cfu/g of live probiotic bacteria to be effective. A decrease in the number of probiotic bacteria may occur throughout the gastrointestinal tract, especially due to stomach acidity and bile. To obtain reliable and accurate data regarding microorganism viability determination in gastrointestinal conditions, in vivo studies on humans are the most ideal. However, clinical studies in humans are not frequently technically, financially, and ethically feasible and are difficult owing to low reproducibility due to individual differences. Moreover, animal models are widely used in in vivo studies. However, their use is avoided as much as possible because it involves animal death or surgical approaches wherein cannulas are inserted into the digestive organ to access the gastrointestinal tract contents. Determining the health benefits of foods requires simulated gastrointestinal studies to optimize digestion and absorption behavior. Therefore, in vitro models are becoming more popular for investigating probiotic behavior in the gastrointestinal tract, as they offer flexibility and reproducibility while avoiding some of the challenges associated with in vivo and animal studies. The positive effects of probiotic microorganisms noted in kefir on health have been known for a long time. Preserving the viability and activity of probiotic microorganisms throughout the production, transportation, storage, and shelf life of kefir is significant. Additionally, these microorganisms must reach the colon by preserving their viability against adverse environmental conditions throughout the human gastrointestinal tract. In this study, an in vitro dynamic model was designed to simulate the human gastrointestinal system, including the mouth, stomach, and small intestine stages. By avoiding the technical difficulties and ethical restrictions of in vivo studies, the number of lactic acid bacteria detected in different kefir samples (IK, SCK, and KG) and the number of viable microorganisms following passage through the in vitro dynamic gastrointestinal tract were determined in a repeatable manner.

Ustun-Aytekin et al. [24] reported that the gastrointestinal digestion of kefir resulted in a significant increase in various bioactive compounds. This suggests that the digestive process releases or enhances the availability of phenolic compounds in kefir. DPPH, which is a measure of antioxidant activity, increased from 4.20% in undigested kefir to 63.06% in digested kefir. This indicates that after undergoing gastrointestinal digestion, kefir becomes a more potent antioxidant. The present study shows that the antioxidant activity of kefir, as measured by DPPH values, can vary in different gastrointestinal tract regions and over different time intervals, particularly in the stomach and small intestine. These variations may be influenced by factors, including the type of kefir and the duration of digestion. Saliva and kefir are the only substances detected in the mouth sample. Over time, secretions from the stomach and small intestine regions (stomach and small intestine buffer solution, respectively) are added. The amounts of HCl and NaOH added to the system to regulate the pH level also caused dilution in the samples collected from these sections. Moreover, it is considered that bioactive compounds with antioxidant effects undergo chemical changes as environmental conditions change. It is believed that this is the cause of the decline in DPPH values. Pasteurized cow milk of the same brand and batch number was used to produce the SCK and KG kefir samples. As shown in Figure 1, kefir produced using KG has more antioxidant activity than kefir produced using SCK. It has been suggested that this is because the microorganisms in the production of SCK and KG kefir differ with regard to diversity.

Vamanu [25] evaluated the resistance of six different probiotic strains (*Weissella paramesenteroides* FT1a, *Lactobacillus* sp. 34.1, *Lactobacillus rhamnosus* [*L. rhamnosus*] E 4.2, *Lactobacillus* sp. 18.1, *Lactobacillus fermentum* [*L. fermentum*] 428ST, and *Lactobacillus plantarum* [*L. plantarum*]) to passage through the gastrointestinal simulator. After 1 h in the stomach, a significant decrease in cell viability was noted for most strains, particularly for the controls (*L. plantarum* 5s and *Escherichia coli* CBAB 2). Most strains showed the most significant decrease in viability after 2–3 h of digestion at the duodenal level. *Lactobacillus* sp. 18.1 experienced the greatest loss. *L. fermentum* 428ST showed relatively high viability during passage, whereas other strains showed varying degrees of viability loss. The study highlighted the significance of selecting probiotic strains that can withstand the challenges of in vitro gastrointestinal transit and the role of certain metabolic factors in their survival. Further, it proposed avenues for future research in this area. Marteau et al. [26] investigated two fermented milk products, each containing different types of microorganisms. The Ofilus^®^ product contained *Bifidobacterium bifidum* and *Lactobacillus acidophilus* [*L. acidophilus*], whereas the yogurt contained *Lactobacillus delbrueckii* [*L. delbrueckii*] ssp. *bulgaricus* and *Streptococcus thermophilus* [*S. thermophilus*] microorganisms. They evaluated the survival of microorganism species in different conditions, such as low- and high-bile concentrations in the stomach, gastric juice, and ileum. The stomach compartment could support *S. thermophilus* and *L. bulgaricus* for only a brief period in the yogurt product. Viability after 40 min was significantly lower than that of *L. acidophilus* and *B. bifidum* in the Ofilus^®^ product; within 70 and 110 min, the viable counts decreased to <1% of the number of consumed bacteria. More than 40% of *L. acidophilus* and *B. bifidum* that had been consumed remained alive in the stomach compartment after 120 min. The cumulative survival rate of *B. bifidum* and *L. acidophilus* in the model was similar to that observed in previous human studies and provides insights into how different factors affect bacterial viability. Similar to our study, differences in survival rates between products at different stages of digestion and under various conditions were highlighted. Regardless of the kefir types used in this study (IK, SCK, or KG), a decrease in the viability of *Lactobacillus* and *Lactococcus* was observed from the mouth region to the last hour of the small intestine region. However, the difference between the values arises from the diverse microorganisms used in kefir production. Briefly, the minimum level of 10^7^ cfu/mL of probiotic bacteria must be present when reaching the intestines to provide a benefit, as determined by various worldwide food organizations and the Turkish Food Codex.

Sharp et al. [27] reported that *Lactobacillus casei* [*L. casei*] 334e in yogurt (pH, 2.0) had lower resistance to acid stress (pH, 2.0) than that detected in cheese. They explained that the possible reason for this situation is that because yogurt has a lower pH level (pH, 4.3), the possibility of sublethal damage to *L. casei* 334e during storage in yogurt may be higher than that in low-fat cheese (pH, 5.1). The number of viable *L. casei* supplements in yogurt decreased from approximately 10^7^ to <10^1^ cfu/g upon exposure to a pH level of 2.0 for 30 min, whereas the number of *L. casei* 334e in low-fat cheese decreased to only approximately 10^5^ cfu/g over 30 min. They determined that when the same exposure was continued, it remained at approximately 10^4^ cfu/g after 120 min. The average initial pH values of the IK products used in the present study were approximately 4.33 + 0.92 (432 [minimum]–4.44 [maximum]). Differences were detected in the average initial pH values of SCK and KG products (5.25 + 0.51 [min–max, 4.89–5.25] and 4.52 + 0.35 [min–max, 4.17–5.13], respectively). Regardless of the initial pH values of the kefir products used, the initial numbers of live *Lactobacillus* and *Lactococcus* in the products were similar. A low pH level is one of the main factors affecting probiotic microorganisms’ growth and survival during kefir storage as well as transportation through the gastrointestinal tract. The quantity of undissociated organic acids in fermented food increases at very low pH values, thereby enhancing the bactericidal effect of these acids. The tolerance to low pH levels varies by strain. The research findings indicate that there was a higher decline in the number of colonies of live *Lactobacillus* between the mouth and stomach regions in the KG and SCK samples (from 9.84 to 7.08 and from 10.35 to 8.69, respectively) than in the IK group (from 9.98 to 9.12). Similar variations were noted in the number of live *Lactococcus*, which may be attributed to the various strains used for the kefir sample production (Table 4 and Table 5).

Comak-Gocer et al. [28] reported that the reduction in viable *L. acidophilus* numbers was significantly affected by the type of dairy products, gastrointestinal model conditions, and storage time. They determined that the highest decrease in *L. acidophilus* numbers was in the yogurt sample, followed by fermented acidophilus milk, the cheese with white pickle, and the ice cream sample. Considering the viability of the two *L. acidophilus* strains in the samples, they reported that *L. acidophilus* ice cream samples were more resistant to the harsh conditions of the dynamic in vitro gastrointestinal model; however, *L. acidophilus* yogurt samples were less resistant to the same model conditions. Moreover, they reported that in the yogurt sample, a decrease was observed for *L. delbrueckii* subsp. *bulgaricus*; however, the decrease in the number of *S. thermophilus* was more pronounced and the number of viable *S. thermophilus* cells could not be determined 180 min after passage through the dynamic in vitro gastrointestinal model. Therefore, they stated that the *L. delbrueckii* subsp. *bulgaricus* strain used in yogurt production can be more resistant to the harsh conditions of the dynamic in vitro gastrointestinal model than the *S. thermophilus* strain used in yogurt production.

Vinderola and Reinheimer [29] evaluated the probiotic properties and biological resistance of 24 lactic acid starter cultures, 24 probiotic bacterial strains, and the barriers (gastric juice and bile salts). They reported that among the probiotic bacteria examined, *L. acidophilus* showed high resistance to gastric juice and bile. Conversely, they determined that among the starter species evaluated, the *L. delbrueckii* subsp. *bulgaricus* was the lactic acid starter culture with the best probiotic properties. It is resistant to gastric juice and bile. Furthermore, they reported that the presence of bile salts was more inhibitory to lactic acid starter bacteria than to probiotic organisms. In their study, *S. thermophilus* was identified as the most sensitive species among the bacteria in the first group, as 0.5% bile salts inhibited most strains. All probiotic bacterial strains were generally more or less resistant to 1% bile salts. Again, when evaluating lactic acid starter cultures in their study, they reported that they showed less resistance to simulated gastric juice than probiotic bacteria at both pH values. All strains in the first group (except *Lb. delbrueckii* subsp. *bulgaricus* Eb4 and *Lc. lactis* Mo12) experienced at least a 6.0-log decrease at a pH level of 2.0, whereas at a pH level of 3.0, the highest decrease in cell viability was detected in *S. thermophilus*. They reported that this decrease was followed by *Lc. lactis* and *Lb. delbrueckii* subsp. *bulgaricus*. *Lb. acidophilus* was the most resistant species among probiotic bacteria, experiencing a 3.4–5.0-log decrease and a 0.7–3.3-log loss in cell viability at pH levels of 2.0 and 3.0, respectively. For bifidobacteria, these values ranged from a 3.3- to 6.0-log decrease at a pH level of 2.0 and from a 0.8- to 2.3-log decrease at a pH level of 3.0, whereas *Lb. casei* and *Lb. rhamnosus* strains experienced a more than 5.0-log decrease at a pH level of 2.0 and a 2.7–5.9-log decrease at a pH level of 3.0. Similarly, in the present study, decreases in the number of lactic acid bacteria were detected, owing to pH level changes. Notably, the total probiotic value of fermented dairy products must consider the gastric and small intestine buffer solutions used in the simulation as well as the probiotic contribution of the lactic acid-initiating microflora in the human body.

Lo Curto et al. [30] reported that following simulated dynamic gastric and duodenal digestion using water as the food matrix, *L. acidophilus johnsonii* (cfu/mL) increased in the logarithmic phase from the first hour to the second hour in the duodenal region and decreased after 2 h of duodenal digestion. They detected a decrease in the logarithmic phase after 1 and 2 h of duodenal digestion when water was used as the food matrix in the intake of *L. casei Shirota* (cfu/mL) and L. *casei immunitas* (cfu/mL). Moreover, they noted that when water was used as the food matrix, survival in the logarithmic phase was significantly related to the pH values of the water. When ingested with milk, it was observed that identical probiotic strains had no effect on the survival rate, pH level, or strain data. They reported that some differences were noted between the three strains they used in their study and that, in general, higher survival rates were observed in milk than in water. Additionally, they reported that this observation could be attributed to the lower buffering capacity of water than that of milk. They stated that the buffering effect of milk could protect the strains from the harmful effects of the stomach and duodenal environments. As lactic acid production is a significant indicator of the adaptation of bacteria that secrete lactose as the product of fermentation, the survival of probiotic strains was confirmed by data obtained by measuring lactic acid production. Another study demonstrated that good survival rates were observed using *Lactobacilli* as carrier vehicles during a simulated in vivo digestion of six *L. rhamnosus* strains. These six probiotic *L. rhamnosus* strains were investigated for their ability to survive in the human upper gastrointestinal tract via a dynamic gastric digestion model. MRS broth was used as the delivery vehicle, and survival levels were investigated during in vitro gastric and gastric plus duodenal digestion. A significant difference was observed in the pH values at the end of gastric digestion. The decrease in the number of bacteria was generally observed more clearly at the end of stomach digestion when the pH level dropped below 2.5. The results highlighted that all tested strains showed a good proportion of viable cells during gastric and duodenal digestion. Consistent with the data, high lactic acid production was detected for all strains, indicating their metabolic activity during digestion [31]. Mainville et al. [4] examined the digestive properties of the in vitro gastrointestinal tract and the upper gastrointestinal tract by incorporating the food matrix. The model evaluated the viability status of bacteria isolated from humans, animals, and fermented milk products. They observed that the viability levels significantly increased for some strains in the dynamic model. Good viability was observed for two strains (*Bifidobacterium animalis* ATCC 25,527 and *Lactobacillus johnsonii* La-1 NCC 533) using both methods. The dynamic model has been shown to better represent events during upper gastrointestinal tract transit than traditional methods by incorporating the food matrix to buffer gastric acidity, thereby exposing bacteria to pH levels noted in vivo before, during, and after the meal. The results of the present study suggested that the consumption of kefir products instead of externally taken probiotic strain supplements is a better way to buffer stomach acidity. However, this should be supported by new studies. Klindt-Toldam et al. [32] compared the viability of a *Lb. acidophilus* NCFM^®^ and *B. lactis* HN019 probiotic mixture in milk chocolate and 72% dark chocolate at different probiotic concentrations during experiments in an in vitro static model of the upper gastrointestinal tract. Plate counts showed a significant decrease (*p* < 0.05) in *Lactobacillus* viability. For all chocolate samples, the viability of *Lb. acidophilus* NCFM^®^ and *B. lactis* HN019 decreased less during simulated digestion. Overall, the survival of both strains during the 65-min digestion was acceptable and relatively parallel across all probiotic concentrations regardless of chocolate type, with one exception determined to be above 6.5 log cfu/g. In milk chocolate (probiotic concentration 2 × 10^8^ cfu/g), *Lb. acidophilus* NCFM^®^ could maintain viability for only up to 35 min; at the end of this period, a 3-log decrease was detected when the pH level dropped below 2.0. However, the same effect was not observed for *B. lactis* HN019, and they reported that this was probably because of the acid-resistant properties of *B. lactis* HN019. The milk contained in milk chocolate has a higher buffering capacity than that of dark chocolate. Jensen et al. [33] conducted studies on *L. plantarum*, *Lactobacillus pentosus*, *Lactobacillus farciminis*, *Lactobacillus sakei*, *Lactobacillus gasseri*, *L. rhamnosus*, *Lactobacillus reuteri*, and *Pediococcus pentosaceus* strains. As a result of the viable cell count following simulated gastric transit tolerance, *L. reuteri* strains and *P. pentosaceus* tolerated gastric juice well and no decrease in their viability was noted, whereas *L. pentosus*, *L. farciminis*, and *L. sakei* strains lost their viability after 180 min. All tested strains tolerated simulated small intestine water, and all strains did not show a decrease in viability against pancreatin and bile enzymes exceeding 1 log. Briefly, in vitro evaluations of potential probiotic bacteria have traditionally paid particular attention to tolerance of the adverse environment of the stomach and small intestine and the ability to adhere to intestinal surfaces. Generally, the ultimate performance criterion of a probiotic is the health benefits it provides to the host. Considering these studies, it has been observed that kefir, which is a dairy product, is less affected by the negative conditions of the stomach and intestines owing to its buffering effect, regardless of the production method. Although a decrease was observed in the initial number of live microorganisms in the products, it was determined that enough live microorganisms reached the colon for probiotic activity.

When five different IK products were evaluated, it was determined that one product was statistically inconsistent. Accordingly, to provide written statistical information, the relevant variable was not included in the statistical analysis evaluation. Furthermore, during the study, the maximum number of kefir varieties produced both industrially and with starter cultures on the market was reached. At the point where the hypotheses were tested, the correlation and regression analyses, which were determined as additional hypotheses, could not be analyzed in this amount of product. The fact that *Lactobacillus* and *Lactococcus* genera were detected in the kefir samples included in the study, but not the substrains, constitutes a limitation regarding interpreting the difference in resistance to environmental conditions.

## 5. Conclusions

The analysis showed statistically significant differences in the dry matter, protein, titration acidity, and °Brix values among the kefir types. However, no significant differences were observed in the ash and lipid content of kefir samples. This study observed no statistically significant difference in DPPH values among the different kefir types, except for a significant difference in the stomach region of kefir produced using KG. *Lactobacillus* and *Lactococcus* are types of lactic acid bacteria commonly detected in fermented products, such as kefir. The analysis showed no significant difference in *Lactobacillus* values among the different kefir types in the mouth, stomach, and small intestine regions. However, in the kefir produced using KG, a statistically significant difference was noted between *Lactobacillus* values in the mouth and stomach regions, with higher values in the mouth. Moreover, a significant difference was noted in the *Lactococcus* values in the stomach region of IK, with higher values at 0 min than those at 120 min. Overall, the results suggest some differences in composition and antioxidant activity between the different kefir types; however, the *Lactobacillus* and *Lactococcus* populations in kefir do not significantly differ between species in various digestive tract regions. However, these findings must be evaluated in the context of the specific experiment and research methodology. The results may be supported by new research on this subject.

## Figures and Tables

**Figure 1 nutrients-15-04808-f001:**
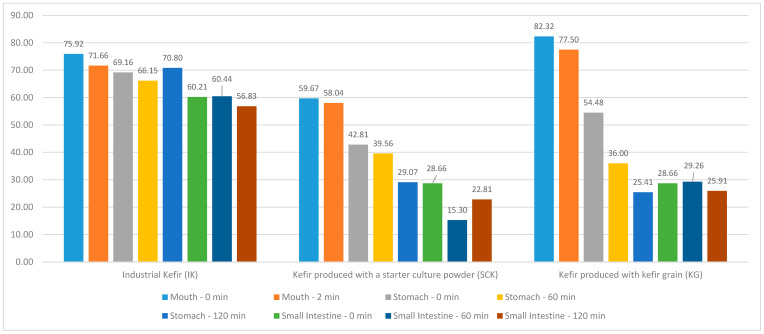
Distribution of DPPH values of IK, SCK, and KG group kefir in mouth region at 0–2 min and in the stomach and small intestine regions at 0, 60, and 120 min.

**Figure 2 nutrients-15-04808-f002:**
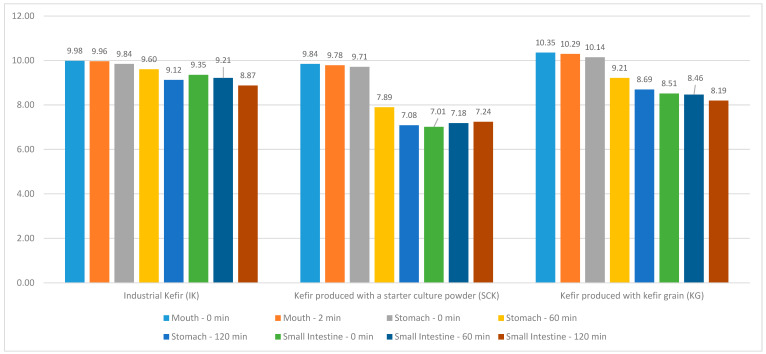
Distribution of Lactobacillus values of IK, SCK, and KG group kefir in mouth region at 0–2 min and in the stomach and small intestine regions at 0, 60, and 120 min.

**Figure 3 nutrients-15-04808-f003:**
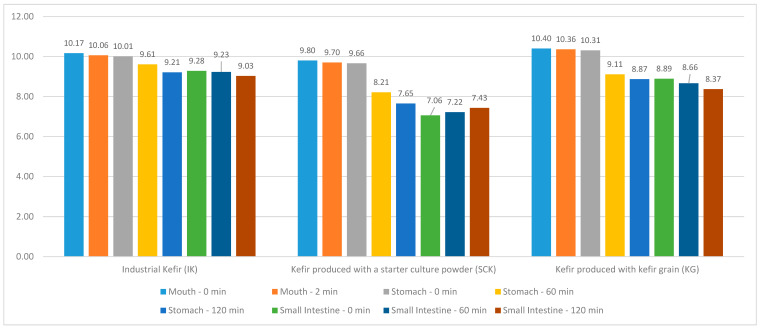
Distribution of *Lactococcus* values of IK, SCK, and KG group kefir in mouth region at 0–2 min and in the stomach and small intestine regions at 0, 60, and 120 min.

**Table 1 nutrients-15-04808-t001:** Comparison of kefir analysis values according to the IK, SCK, and KG groups.

	Group	$\overset{-}{X}$ ± SD	Median (Min–Max)	F-H	*p*
Dry matter (%)	IK	10.05 ± 0.58	10.15 ^a^ (8.74–10.73)	H = 29.988	<0.001 ***
	SCK	11.26 ± 0.16	11.27 ^b^ (11.06–11.44)		
	KG	11.30 ± 0.16	11.28 ^b^ (11.02–11.62)		
Ash (%)	IK	0.62 ± 0.12	0.63 (0.33–0.84)	F = 1.081	0.349
	SCK	0.66 ± 0.03	0.66 (0.60–0.71)		
	KG	0.66 ± 0.05	0.67 (0.56–0.74)		
Lipid (%)	IK	2.67 ± 0.28	2.70 (2.20–3.00)	H = 4.890	0.087
	SCK	2.89 ± 0.10	2.85 (2.80–3.00)		
	KG	2.87 ± 0.08	2.90 (2.70–3.00)		
Protein (%)	IK	2.78 ± 0.15	2.79 ^c^ (2.54–3.01)	H = 26.840	<0.001 ***
	SCK	2.39 ± 0.39	2.32 ^b^ (2.01–2.90)		
	KG	2.02 ± 0.32	2.01 ^a^ (1.55–2.45)		
Titration acidity (% Lactic acid)	IK	0.82 ± 0.06	0.81 ^b^ (0.74–0.92)	H = 18.058	<0.001 ***
	SCK	0.88 ± 0.05	0.87 ^c^ (0.81–0.96)		
	KG	0.73 ± 0.09	0.76 ^a^ (0.58–0.86)		
Soluble Solids Content (°Brix)	IK	7.77 ± 0.47	8.00 ^a^ (7.00–8.20)	H = 32.583	<0.001 ***
	SCK	8.83 ± 0.17	8.80 ^c^ (8.60–9.00)		
	KG	8.40 ± 0.25	8.40 ^b^ (8.00–9.00)		

F: one-way ANOVA test; H: Kruskal–Wallis H test; *** *p* < 0.001; ^a, b, c^: the difference between medians that do not have a common letter is significant (*p* > 0.05).

**Table 2 nutrients-15-04808-t002:** Comparison of IK, SCK, and KG data with DPPH values at the mouth, stomach, and small intestine regions.

	IK		SCK		KG		H	p2
	$\overset{-}{X}$ ± SD	Median (Min–Max)	$\overset{-}{X}$ ± SD	Median (Min–Max)	$\overset{-}{X}$ ± SD	Median (Min–Max)		
Mouth—0 min	75.92 ± 9.15	76.11 (66.02–85.44)	59.67 ± 30.25	59.67 (38.28–81.06)	82.32 ± 19.12	89.77 (48.47–94.59)	3.818	0.148
Mouth—2 min	71.66 ± 6.90	73.92 (61.58–77.22)	58.04 ± 29.25	58.04 (37.35–78.72)	77.50 ± 11.20	75.11 (61.95–90.80)	1.641	0.440
T-W	T = 1.611		W = −1.342		W = −0.944			
p1	0.205		0.180		0.345			
Stomach—0 min	69.16 ± 8.89	71.26 (56.67–77.46)	42.81 ± 8.06	42.81 (37.11–48.51)	54.48 ± 15.18 ^b^	53.95 (35.10–74.57)	4.200	0.122
Stomach—60 min	66.15 ± 5.20	67.11 ^B^ (59.05–71.31)	39.56 ± 3.11	39.56 ^AB^ (37.36–41.76)	36.00 ± 6.11 ^b^	38.85 ^A^ (28.36–42.67)	7.064	0.029 *
Stomach—120 min	70.80 ± 2.92	71.26 ^B^ (67.18–73.49)	29.07 ± 1.38	29.07 ^AB^ (28.09–30.04)	25.41 ± 10.07 ^a^	26.35 ^A^ (9.02–34.81)	7.064	0.029 *
R-FD	R = 0.578		FD = 3.000		R = 7.765			
p1	0.503		0.223		0.013 *			
S. Intestine–0 min	60.21 ± 9.10	59.11 ^B^ (51.64–70.97)	28.66 ± 1.83	28.66 ^A^ (27.36–29.95)	28.66 ± 8.18	29.63 ^A^ (16.64–39.50)	7.000	0.030 *
S. Intestine—60 min	60.44 ± 5.12	59.55 ^B^ (55.34–67.32)	15.30 ± 0.08	15.30 ^A^ (15.25–15.36)	29.26 ± 6.30	26.78 ^A^ (22.67–36.57)	8.591	0.014 *
S. Intestine—120 min	56.83 ± 5.86	56.85 ^B^ (51.65–61.97)	22.81 ± 12.21	22.81 ^A^ (14.17–31.45)	25.91 ± 9.68	22.72 ^A^ (14.61–37.16)	7.255	0.027 *
R-FD	FD = 1.500		FD = 1.000		R = 0.264			
p1	0.472		0.607		0.775			

T: dependent sample *t*-test; W: Wilcoxon signed-rank test; R: repeated measures ANOVA test; FD: Friedman test; H: Kruskal–Wallis H test; p1: comparison is made between the values of 0–2 min, 0–60 min, and 120 min within the group; p2: comparison is made between the groups at 0, 2, 60, and 120 min; * *p* < 0.05: ^a, b, A, B^: the difference between the mean and medians that do not have a common letter is significant (*p* > 0.05).

**Table 3 nutrients-15-04808-t003:** Comparison of DPPH, *Lactobacillus*, and *Lactococcus* difference values of the IK, SCK, and KG groups in the mouth, stomach, and small intestine regions.

		IK		SCK		KG			
		$\overset{-}{X}$ ± SD	Median (Min–Max)	$\overset{-}{X}$ ± SD	Median (Min–Max)	$\overset{-}{X}$ ± SD	Median (Min–Max)	H	p2
DPPH	Mouth—Difference	−4.26 ± 5.29	−6.05 (−8.22–3.27)	−1.63 ± 1.00	−1.64 (−2.34–−0.93)	−4.81 ± 20.01	−4.17 (−29.84–25.59)	1.391	0.499
	Stomach—Difference	1.63 ± 7.96	−0.93 ^A^ (−4.66–13.04)	−13.75 ± 6.68	−13.75 ^AB^ (−18.47–−9.02)	−29.06 ± 21.82	−21.32 ^B^ (−65.55–−8.75)	7.573	0.023 *
	S. Intestine—Difference	−3.38 ± 12.13	−2.19 (−19.32–10.19)	−5.85 ± 14.05	−5.85 (−15.78–4.09)	−2.76 ± 8.54	−2.34 (−12.69–6.08)	0.255	0.880
	F-H	F = 0.508		H = 2.000		F = 3.381			
	p1	0.618		0.368		0.068			
*Lactobacillus*	Mouth—Difference	−0.03 ± 0.04 ^b^	−0.04 (−0.06–0.03)	−0.05 ± 0.02	−0.05 (−0.06–−0.04)	−0.06 ± 0.03 ^b^	−0.06 (−0.11–−0.02)	2.914	0.233
	Stomach—Difference	−0.72 ± 0.47 ^a^	−0.72 (−1.30–−0.16)	−2.63 ± 0.57	−2.63 (−3.03–−2.22)	−1.44 ± 0.59 ^a^	−1.18 (−2.37–−0.93)	5.823	0.054
	S. Intestine—Difference	−0.48 ± 0.65 ^ab^	−0.37 (−1.31–0.12)	0.23 ± 0.27	0.23 (0.04–0.42)	−0.32 ± 1.15 ^ab^	−0.06 (−2.23–0.60)	1.618	0.445
	F-H	F = 2.354		H = 4.571		F = 4.840			
	p1	0.151		0.102		0.029 *			
*Lactococcus*	Mouth—Difference	−0.11 ± 0.02	−0.11 (−0.14–−0.09)	−0.09 ± 0.04	−0.09 (−0.12–−0.06)	−0.04 ± 0.03 ^b^	−0.03 (−0.07–−0.01)	6.005	0.050
	Stomach—Difference	−0.80 ± 0.35	−0.77 (−1.25–−0.42)	−2.01 ± 0.72	−2.01 (−2.51–−1.50)	−1.44 ± 0.64 ^a^	−1.28 (−2.40–−0.68)	5.268	0.072
	S. Intestine—Difference	−0.25 ± 0.52	−0.07 (−1.01–0.14)	0.37 ± 0.06	0.37 (0.33–0.42)	−0.52 ± 1.19 ^ab^	−0.09 (−2.58–0.35)	3.641	0.162
	F-H	F = 4.038		H = 4.571		F = 4.151			
	p1	0.056		0.102		0.043 *			

F: one-way ANOVA test; H: Kruskal–Wallis H test; p1: intragroup difference values in the mouth, stomach, and intestine regions are compared; p2: the difference values in the mouth, stomach, and intestine regions between the groups are compared; * *p* < 0.05, ^a, b^: the difference between medians that do not have a common letter is significant (*p* < 0.05); ^A, B^: the difference between medians that do not have a common letter is significant (*p* < 0.05).

**Table 4 nutrients-15-04808-t004:** Comparison of IK, SCK, and KG data of *Lactobacillus* values in the mouth, stomach, and small intestine regions.

	IK		SCK		KG			
	$\overset{-}{X}$ ± SD	Median (Min–Max)	$\overset{-}{X}$ ± SD	Median (Min–Max)	$\overset{-}{X}$ ± SD	Median (Min–Max)	H	p2
Mouth—0 min	9.98 ± 0.27	9.91 (9.75–10.36)	9.84 ± 0.46	9.84 (9.51–10.16)	10.35 ± 0.08	10.38 (10.21–10.41)	5.823	0.054
Mouth—2 min	9.96 ± 0.24	9.87 (9.78–10.31)	9.78 ± 0.47	9.78 (9.45–10.12)	10.29 ± 0.08	10.30 (10.15–10.37)	5.073	0.079
T-W	T = 1.380		W = −1.342		T = 3.968			
p1	0.261		0.180		0.017 *			
Stomach—0 min	9.84 ± 0.22	9.76 (9.68–10.16)	9.71 ± 0.46	9.71 (9.38–10.04)	10.14 ± 0.17 ^b^	10.21 (9.91–10.29)	4.041	0.133
Stomach—60 min	9.60 ± 0.10	9.62 (9.47–9.69)	7.89 ± 1.66	7.89 (6.72–9.06)	9.21 ± 0.50 ^a^	9.11 (8.81–10.07)	4.664	0.097
Stomach—120 min	9.12 ± 0.46	9.23 (8.48–9.51)	7.08 ± 1.04	7.08 (6.35–7.82)	8.69 ± 0.47 ^a^	8.85 (7.92–9.07)	5.973	0.050
R-FD	R = 5.719		FD = 4.000		R = 11.146			
p1	0.059		0.135		0.019*			
S. Intestine—0 min	9.35 ± 0.34	9.41 ^B^ (8.90–9.69)	7.01 ± 1.27	7.01 ^A^ (6.11–7.90)	8.51 ± 0.45	8.62 ^A^ (7.74–8.95)	7.205	0.027 *
S. Intestine—60 min	9.21 ± 0.18	9.23 (8.99–9.38)	7.18 ± 1.18	7.18 (6.35–8.01)	8.46 ± 0.73	8.59 (7.57–9.37)	5.268	0.072
S. Intestine—120 min	8.87 ± 0.37	8.93 (8.38–9.25)	7.24 ± 1.00	7.24 (6.53–7.94)	8.19 ± 0.91	8.26 (6.71–9.18)	5.255	0.072
R-FD	R = 2.284		FD = 3.000		0.401			
p1	0.227		0.223		0.606			

T: dependent sample *t*-test; W: Wilcoxon signed rank test; R: repeated measures ANOVA test; FD: Friedman test; H: Kruskal–Wallis H test; p1: a comparison is made between the values of 0–2 min, and 0, 60, and 120 min within the group; p2: comparison is made between the groups at 0, 2, 60, and 120 min; * *p* < 0.05; ^a, b, A, B^: the difference between means and medians that do not have a common letter is significant (*p* > 0.05).

**Table 5 nutrients-15-04808-t005:** Comparison of IK, SCK, and KG data of *Lactococcus* values in the mouth, stomach, and small intestine regions.

	IK		SCK		KG			
	$\overset{-}{X}$ ± SD	Median (Min–Max)	$\overset{-}{X}$ ± SD	Median (Min–Max)	$\overset{-}{X}$ ± SD	Median (Min–Max)	H	p2
Mouth—0 min	10.17 ± 0.22	10.18 ^AB^ (9.96–10.38)	9.80 ± 0.02	9.80 ^A^ (9.78–9.81)	10.40 ± 0.07	10.43 ^B^ (10.30–10.48)	6.505	0.039 *
Mouth—2 min	10.06 ± 0.23	10.06 ^A^ (9.85–10.26)	9.70 ± 0.02	9.70 ^A^ (9.69–9.72)	10.36 ± 0.06	10.37 ^B^ (10.27–10.42)	8.591	0.014 *
T-W	W = −1.826		W = −1.342		T = 3.101			
p1	0.068		0.180		0.036 *			
Stomach—0 min	10.01 ± 0.27 ^b^	10.04 ^A^ (9.72–10.25)	9.66 ± 0.03	9.66 ^A^ (9.64–9.68)	10.31 ± 0.04	10.32 ^B^ (10.25–10.37)	7.814	0.020 *
Stomach—60 min	9.61 ± 0.33 ^ab^	9.64 (9.20–9.97)	8.21 ± 1.48	8.21 (7.17–9.26)	9.11 ± 0.51	9.13 (8.34–9.77)	3.023	0.221
Stomach—120 min	9.21 ± 0.24 ^a^	9.22 (8.97–9.44)	7.65 ± 0.69	7.65 (7.17–8.14)	8.87 ± 0.65	9.02 (7.92–9.69)	4.041	0.133
R-FD	R = 10.236		FD = 4.000		R = 25.771			
p1	0.017 *		0.135		<0.001 ***			
S. Intestine—0 min	9.28 ± 0.37	9.20 (8.98–9.75)	7.06 ± 0.71	7.06 (6.56–7.56)	8.89 ± 0.65 ^c^	8.91 (8.18–9.58)	5.155	0.076
S. Intestine—60 min	9.23 ± 0.42	9.18 ^B^ (8.83–9.74)	7.22 ± 0.28	7.22 ^A^ (7.02–7.42)	8.66 ± 0.52 ^b^	8.62 ^AB^ (8.23–9.55)	6.505	0.039 *
S. Intestine—120 min	9.03 ± 0.35	8.92 (8.73–9.54)	7.43 ± 0.77	7.43 (6.89–7.98)	8.37 ± 0.94 ^a^	8.43 (6.90–9.49)	5.823	0.054
R-FD	R = 0.744		FD = 1.000		R = 0.929			
p1	0.514		0.607		0.434			

T: dependent sample *t*-test; W: Wilcoxon signed rank test; R: repeated measures ANOVA test; FD: Friedman test; H: Kruskal–Wallis H test; p1: a comparison is made between the values of 0–2 min and 0, 60, and 120 min within the group; p2: a comparison is made between the groups between the values of 0, 2, 60, and 120 min; * *p* < 0.05; *** *p* < 0.001; ^a, b, c, A, B^: the difference between means and medians that do not have a common letter is significant (*p* < 0.05).

## Data Availability

The data obtained in this study are available from the corresponding author upon request.

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
