# Peer review of "Determination of the Viability of Lactic Acid Bacteria by Dynamic In Vitro Gastrointestinal Model in Household and Industrial-Type Kefir Samples"

_nutrients, 2023, doi:10.3390/nu15224808_

Round 1
Reviewer 1 Report
Comments and Suggestions for Authors
This manuscript by Merve Ince Palamutoglu, et al. investigated variations in antioxidant activity, and the counts of lactic acid bacteria as they move through different stages of simulated gastrointestinal digestion in various regions of the digestive tract of kefir samples manufactured using various techniques. The authors showed statistically significant differences in the dry matter, protein, titration acidity, and Brix° values among the kefir types. The study found no statistically significant difference in DPPH values among the different types of kefir, except for a significant difference in the stomach region of kefir produced with kefir grains. In my opinion, the authors of the manuscript have done a good job formulating the study plan and presenting their findings in clear manner. However, the manuscript is missing several important data.
The manuscript can be improved by addressing following issues:
- - Include significance of study findings in the Abstract
- - Define abbreviation DPPH
- - Discussion section is merely a summary of previous studies. Discuss in detail about importance of your study findings and compare it with previously published studies, and how your study findings are unique and important
- - Comment on importance of your study findings for clinical practice
- - Include study limitations
Author Response
Thank you for reviewing and contributing to our manuscript.
- Include significance of study findings in the Abstract
The significance of the study findings was included in the summary.
- Define abbreviation DPPH
The abbreviation DPPH was explained on line 188 as "2,2-diphenyl-1-picrylhydrazyl".
- Discussion section is merely a summary of previous studies. Discuss in detail about importance of your study findings and compare it with previously published studies, and how your study findings are unique and important.
In the discussion section, the interpretation of the findings was updated and compared with previously published studies.
- Comment on importance of your study findings for clinical practice
The importance of the findings for clinical practice was emphasized again in the discussion section.
- Include study limitations
The limitations of the study based were stated in the discussion section.

Reviewer 2 Report
Comments and Suggestions for Authors
After reviewing the manuscript titled “Determination of Viability of Lactic Acid Bacteria by Dynamic In Vitro Gastrointestinal Model in Household and Industrial-Type Kefir Samples”, that aimed to compare the proximate composition of the kefir samples, antioxidant activity, and number of lactic acid bacteria with the initial probiotic character and to examine the changes throughout the simulated gastrointestinal tract, I came to the conclusion, that this manuscript could not be accepted in the current form and needs major revision. First of all, English language editing is required as some parts of the manuscript is really difficult to read and understand. Abstract should be rewritten as even the first sentence of the abstract is not clear. Also, please look through all the names of the microorganisms, all of them should be written in Italic. Thorough text editing is required. In the microbiological analysis section you state that you took 10 g of kefir sample, kefir is a liquid and 10 mL of sample should be taken. Also you state that the number of viable microorganisms was measured. How it was measured? Terminology should be checked.
Comments on the Quality of English Language
Majors English language editing is required as some parts of the manuscript is really difficult to read and understand.
Author Response
Thank you for reviewing and contributing to our manuscript. We added the revisions below.
- English language editing and proof reading was applied.
- In line with your suggestions, the first sentence of the abstract was reviewed, and the sentence was corrected.
"The research provides findings from the variations in their antioxidant activity, and the counts of lactic acid bacteria as they move through different stages of simulated gastrointestinal digestion in various regions of the digestive tract of kefir samples manufactured using various techniques." sentence changed as " This study presents results based on differences in the antioxidant activity and lactic acid bacteria counts in different parts of the digestive tract following simulated gastrointestinal digestion of kefir samples."
- All names of microorganisms were reviewed and the names of microorganisms that were not written in italics by mistake have been italicized. It may be because of the first manuscript that we upload was in italics, but journal format was changed the italics.
- The expression 10 g kefir sample in line 203 was written incorrectly by mistake and was corrected as 10 ml.
- Writing the number of live microorganisms as a measurement was a translation error and was checked. The revision was made (Line 198).
"1 mL samples were taken from the mouth at 0th and 2nd minutes, and at 0th, 60th, and 120th minutes in the stomach and small intestine, and the number of viable microorganisms was measured." was changed as "1-mL samples were taken from the mouth at 0 and 2 min, and from the stomach and small intestine at 0, 60, and 120 min. The number of live microorganisms in the samples was determined as cfu/mL.
- The units of the data in Table 1 have been added.

Round 2
Reviewer 1 Report
Comments and Suggestions for Authors
Authors addressed all of my concerns and incorporated my suggestions in the revised manuscript.
Reviewer 2 Report
Comments and Suggestions for Authors
All comments were addressed properly and necessary improvements were made.